# OFFLINE ROBUSTNESS OF DISTRIBUTIONAL ACTOR-CRITIC ENSEMBLE REINFORCEMENT LEARNING

## ABSTRACT

Offline reinforcement learning (RL) focuses on learning policies using static datasets without further exploration. With the introduction of distributional reinforcement learning into offline RL, current methods excel at quantifying the risk and ensuring the security of learned policies. However, these algorithms cannot effectively balance the distribution shift and robustness, and even a minor perturbation in observations can significantly impair policy performance. In this paper, we propose the algorithm of Offline Robustness of Distributional actor-critic Ensemble Reinforcement learning (ORDER) to improve the robustness of policies. In ORDER, we introduce two approaches to enhance the robustness: i) introduce the smoothing technique to policies and distribution functions for states near the dataset; ii) strengthen the quantile network. In addition to improving the robustness, we also theoretically prove that ORDER converges to a conservative lower bound, which can alleviate the distribution shift. In our experiments, we validate the effectiveness of ORDER in the D4RL benchmark through comparative experiments and ablation studies.

## 1 INTRODUCTION

Offline reinforcement learning (RL) (Lange et al., 2012; Brandfonbrener et al., 2021; Wang et al., 2021; Nguyen-Tang et al., 2023) concerns the problem of learning a policy from a fixed dataset without further interactions. Offline RL can reduce risk and costs since it eliminates the need for online interaction. In this way, offline RL can be well used in real-world applications such as autonomous driving (Diehl et al., 2023), healthcare (Zhang et al., 2023) and robot control (Singh et al., 2022).

Applying the standard policy improvement approaches to an offline dataset typically results in the distribution shift problem, making offline RL a challenging task (Haarnoja et al., 2018; Ashvin et al., 2020). Some prior works have relieved this issue by penalizing the action-value of the out-of-distribution (OOD) actions (Fujimoto et al., 2019; Fujimoto & Gu, 2021; Lyu et al., 2022). Nevertheless, simply learning the expectation of action-value is unable to quantify risk and ensure that the learned policy acts safely. To overcome this problem, some efforts have been made to import distributional RL (Dabney et al., 2018a;b; Ma et al., 2020) into offline RL to learn the full distribution over future returns, which is used to make plans to avoid risky and unsafe actions. In addition, with the establishment of risk-sensitive objectives (Ma et al., 2020), distributional offline RL (Ma et al., 2021; Bai et al., 2022b) learns state representations better since they can acquire richer distributed signals, making them superior to traditional reinforcement learning algorithms even on risk-seeking and risk-averse objectives.

Unfortunately, research on distributional offline RL is less complete. CODAC (Ma et al., 2021) brings distributional RL into the offline setting by penalizing the predicted return quantiles for OOD actions. Meanwhile, MQN-CQR (Bai et al., 2022b) learns a worst-case policy by optimizing the conditional value-at-risk of the distributional value function. However, existing distributional offline RL methods only focus on the safety of the learned policy. These methods leverage a conservative return distribution to impair the robustness, and will make policies highly sensitive, even a minor perturbation in observations (Kumar et al., 2020). As a result, merely possessing safety can not make a fine balance between conservatism and robustness, which does not pay enough attention to robustness.

In this paper, we propose Offline Robustness of Distributional actor-critic Ensemble Reinforcement Learning (ORDER) by introducing a smoothing technique to quantile networks. Firstly, we consider the dynamic entropy regularizer of the quantile function instead of an unchangeable constant to ensure sufficient exploration. Secondly, the increasing number of quantile networks is also beneficial to obtain a more robust distribution value function. Thirdly, smooth regularization is brought into the distribution and policies of states near the dataset. In theory, we prove that ORDER obtains a uniform lower bound on all integrations of the quantiles with the distribution soft Bellman operator, which controls the distribution shift. Such bound also achieves the same effect for both expected returns and risk-sensitive objectives. Overall, ORDER can mitigate the OOD problem and simultaneously balance conservatism and robustness.

In our experiments, ORDER outperforms the existing distributional offline RL methods in the D4RL benchmark (Fu et al., 2020). Meanwhile, our algorithm is also competitive against the current advanced algorithms. Our ablation experiments demonstrate that strengthening the quantile network is critical to the performance of ORDER. In addition, choosing different risk measure functions does not have a great impact on the performance of ORDER, which also shows the robustness of our method.

## 2 PRELIMINARIES

### 2.1 MARKOV DECISION PROCESS AND OFFLINE REINFORCEMENT LEARNING

Consider an episodic Markov decision process (MDP) $M = (\mathcal{S}, \mathcal{A}, \mathbb{P}, T, R, \gamma)$, where $\mathcal{S}$ is the state space, $\mathcal{A}$ is the action space, $\mathbb{P}(s'|s, a)$ is the transition distribution, $T$ is the length of the episode, $R(s, a)$ is the reward function, and $\gamma$ is the discount factor. For a stochastic policy $\pi(a|s) : \mathcal{S} \times \mathcal{A} \to \mathbb{R}$, action-value function $Q^\pi$ is defined as

$$Q^\pi(s, a) := \mathbb{E}_\pi[\sum_{t=0}^{\infty} \gamma^t R(s_t, a_t)] \mid a_t \sim \pi(\cdot|s_t), s_{t+1} \sim \mathbb{P}(\cdot|s_t, a_t), s_0 = s, a_0 = a.$$

Assuming rewards satisfy $R(s_t, a_t) \in [R_{\min}, R_{\max}]$, then $Q^\pi(s, a) \in [V_{\min}, V_{\max}] \subseteq [R_{\min}/(1 - \gamma), R_{\max}/(1 - \gamma)]$. Standard RL aims at learning the optimal policy $\pi^\star$ such that $Q^{\pi^\star}(s, a) \geq Q^\pi(s, a)$ for all $s \in \mathcal{S}, a \in \mathcal{A}$ and all $\pi$. The corresponding $Q$-function of the policy satisfies the Bellman operator:

$$\mathcal{T}^\pi Q(s, a) = \mathbb{E}[R(s, a)] + \gamma \cdot \mathbb{E}_{\mathbb{P}, \pi}[Q(s', a')].$$

In the offline setting, the agent is not allowed to interact with the environment (Fu et al., 2020). The objective of agents is to learn an optimal policy only from a fixed dataset $\mathcal{D} := \{(s, a, r, s')\}$. For all states $s \in \mathcal{D}$, let $\hat{\pi}(a|s) := \frac{\sum_{s', a' \in \mathcal{D}} \mathbf{1}_{(s'=s, a'=a)}}{\sum_{s' \in \mathcal{D}} \mathbf{1}_{s'=s}}$ be the empirical behavior policy. In order to avoid the situation where the denominator of the fraction is 0 in the theoretical analysis, we assume that $\hat{\pi}(a|s) > 0$. Broadly, actions that are not drawn from $\hat{\pi}$ (i.e., those with low probability density) are the out-of-distribution (OOD).

An approximation $\hat{\mathcal{T}}^\pi$ is used in fitted $Q$-evaluation (FQE) (Riedmiller, 2005; Ernst et al., 2005) that $R$ and $\mathbb{P}$ in $\mathcal{T}^\pi$ are replaced by estimates $\hat{R}$ and $\hat{\mathbb{P}}$ based on $\mathcal{D}$. Next, sampling $(s, a, r, s')$ from $\mathcal{D}$, Bellman target can be estimated as $\hat{\mathcal{T}}^\pi Q(s, a) = \mathbb{E}[\hat{R}(s, a)] + \gamma \mathbb{E}_{\hat{\mathbb{P}}, \pi}[Q(s', a')]$. Since $Q^\pi = \mathcal{T}^\pi Q^\pi$ is the fixed point of the Bellman operator $\mathcal{T}^\pi$, $Q^\pi$ can be evaluated by computing $\hat{Q}^{k+1} := \arg\min_Q \mathcal{L}(\hat{Q}, \hat{\mathcal{T}}^\pi \hat{Q}^k)$, where $\mathcal{L}(Q, Q') = \mathbb{E}_{\mathcal{D}(s, a)}[(Q(s, a) - Q'(s, a))^2]$. If $\hat{\mathcal{T}}^\pi = \mathcal{T}^\pi$, then $\lim_{k \to \infty} \hat{Q}^k = Q^\pi$.

### 2.2 DISTRIBUTIONAL REINFORCEMENT LEARNING

Instead of a scalar $Q^\pi(s, a)$, distributional RL model intrinsic randomness of return by learning distribution action-value $Z^\pi(s, a) = \sum_{t=0}^{\infty} \gamma^t R(s_t, a_t)$ (Bellemare et al., 2017). Obviously, $Q$-function is the expectation of the return distribution, i.e., $Q^\pi(s, a) = \mathbb{E}[Z^\pi(s, a)]$. The distributional Bellman operator for policy evaluation is $\mathcal{T}^\pi Z(s, a) :\overset{D}{=} R(s, a) + \gamma Z(s', a')$, where $\overset{D}{=}$ indicates

equality in distribution. Define the quantile function be the inverse of the cumulative density function $F_Z(z) = Pr(Z < z)$ as $F_Z^{-1}(\tau) := \inf\{z \in \mathbb{R} : \tau \le F_Z(z)\}$ (Müller, 1997), where $\tau$ denotes the quantile fraction. For random variables $U$ and $V$ with quantile functions $F_U^{-1}$ and $F_V^{-1}$, the $p$-Wasserstein distance $W_p(U, V) = (\int_0^1 |F_U^{-1}(\tau) - F_V^{-1}(\tau)|^p d\tau)^{1/p}$ is the $L_p$ metric on quantile functions. Bellemare et al. (2017) gave that the distributional Bellman operator $\mathcal{T}^\pi$ is a $\gamma$-contraction in the $W_p$, i.e., let $\bar{d}_p(Z_1, Z_2) := \sup_{s,a} W_p(Z_1(s, a), Z_2(s, a))$ be the largest Wasserstein distance over $(s, a)$, and $\mathcal{Z} = \{Z : \mathcal{S} \times \mathcal{A} \to \mathcal{P}(\mathbb{R}) | \forall(s, a), \mathbb{E}[|Z(s, a)|^p] < \infty\}$ be the space of distributions over $\mathbb{R}$ with bounded $p$-th moment, then

$$\bar{d}_p(\mathcal{T}^\pi Z_1, \mathcal{T}^\pi Z_2) \le \gamma \cdot \bar{d}_p(Z_1, Z_2) \qquad \forall Z_1, Z_2 \in \mathcal{Z}.$$

Fitted distributional evaluation (FDE) (Ma et al., 2021) approximates $\mathcal{T}^\pi$ by $\hat{\mathcal{T}}^\pi$ using $\mathcal{D}$, then $Z^\pi$ can be estimated by starting from an arbitrary $\hat{Z}^0$ and iteratively computing

$$\hat{Z}^{k+1} = \arg\min_Z \mathcal{L}_p(Z, \hat{\mathcal{T}}^\pi \hat{Z}^k), \quad \text{where} \quad \mathcal{L}_p(Z, Z') = \mathbb{E}_{\mathcal{D}(s,a)}[W_p(Z(s, a), Z'(s, a))^p].$$

In distributional RL, let risk measure function $\Phi : \mathcal{Z} \to \mathbb{R}$ be a map from the value distribution space to real numbers. Given a distorted function $g(\tau)$ over $[0, 1]$, the distorted expectation of $Z$ is

$$\Phi_g(Z(s, a)) = \int_0^1 F_{Z(s,a)}^{-1}(\tau) g(\tau) d\tau,$$

and the corresponding policy is $\pi_g(s) := \arg\max_a \Phi_g(Z(s, a))$ (Dabney et al., 2018b). Specially, if $g = \text{Uniform}([0, 1])$, then $Q^\pi(s, a) = \Phi_g(Z(s, a))$. For other choices of $g(\tau)$, please refer to Section 5.2.

### 2.3 ROBUST REINFORCEMENT LEARNING

Robust RL learns the policy by introducing worst-case adversarial noise to the system dynamics and formulating the noise distribution as the solution of a zero-sum minimax game. In order to learn robust policy $\pi$, SR$^2$L (Shen et al., 2020) obtains $\hat{s} = \arg\max_{\hat{s} \in \mathbb{B}_d(s,\epsilon)} D_J(\pi(\cdot|s) || \pi(\cdot|\hat{s}))$ by adding a perturbation to the state $s$, where $\mathbb{B}_d(s, \epsilon) = \{\hat{s} : d(s, \hat{s}) \le \epsilon\}$ and the Jeffrey's divergence $D_J$ for two distributions $P$ and $Q$ is defined by $D_J(P||Q) = \frac{1}{2}[D_{\text{KL}}(P||Q) + D_{\text{KL}}(Q||P)]$, and then define a smoothness regularizer for policy as $\mathcal{R}_s^\pi = \mathbb{E}_{s \sim \rho^\pi} \max_{\hat{s} \in \mathbb{B}_d(s,\epsilon)} D_J(\pi(\cdot|s) || \pi(\cdot|\hat{s}))$. Analogously, $\mathcal{R}_s^Q = \mathbb{E}_{s \sim \rho^\pi} \max_{\hat{s} \in \mathbb{B}_d(s,\epsilon)} (Q(s, a) - Q(\hat{s}, a))^2$ is the smoothness regularizer for $Q$-function, where $\rho^\pi$ is state distribution.

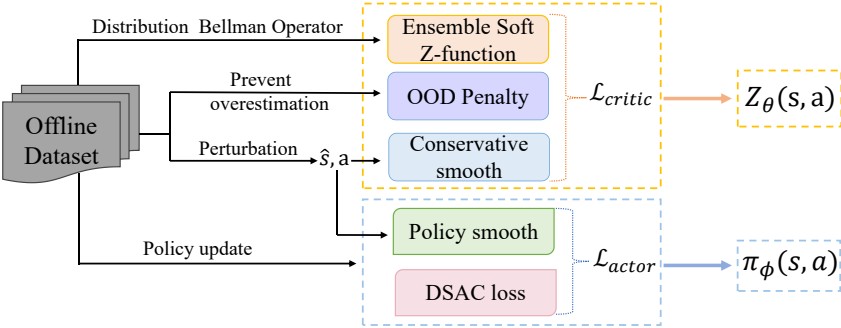

Figure 1: Architecture diagram for ORDER.

## 3 OFFLINE ROBUSTNESS OF DISTRIBUTIONAL ENSEMBLE ACTOR-CRITIC

In ORDER, we first obtain a state with adversarial perturbations, and then introduce the smoothness regularization to both the policy and the distribution action-value function for states with adversarial noises. The smooth regularization can be used to learn a smooth $Z$-function and generate a

smooth policy, which makes the algorithm robust. However, the introduction of smoothness could potentially result in an overestimation of values at the boundaries of the supported dataset. To overcome this problem, we incorporate a penalty factor for OOD actions to reduce the quantile values of these actions. In addition, we strengthen the quantile network by increasing the number of quantile networks, which is also beneficial to the robustness of our algorithm. The overall architecture of ORDER is shown in Fig. 1.

## 3.1 ROBUST DISTRIBUTIONAL ACTION-VALUE FUNCTION

In this part, we sample three sets of state-action pairs and form three different loss functions to obtain a conservative smooth policy. First of all, we construct a perturbation set $\mathbb{B}_d(s, \epsilon)$ to obtain $(\hat{s}, a)$ pairs, where $\mathbb{B}_d(s, \epsilon)$ is an $\epsilon$-radius ball measured in metric $d(\cdot, \cdot)$ and $\hat{s} \in \mathbb{B}_d(s, \epsilon)$. Then we sample $(s, \hat{a})$ pairs from the current policy $\pi_\phi$, where $\hat{a} \sim \pi_\phi(s)$. ORDER contains $M$ $Z$-function and denotes the parameters of the $m$-th $Z$-function and the target $Z$-function as $\theta_m$ and $\theta_m'$, respectively. With the help of the constructions, we will give the different learning targets for $(s, a)$, $(\hat{s}, a)$ and $(s, \hat{a})$ pairs, respectively.

In DSAC (Ma et al., 2020), based on the idea of maximum entropy RL (Haarnoja et al., 2018), the distribution soft Bellman operator is defined as

$$\mathcal{T}_{\mathrm{DS}}^\pi Z(s, a) \overset{D}{:=} R(s, a) + \gamma[Z(s', a') - c\log\pi(a'|s')].$$

The algorithm based on maximum entropy RL has better robustness and stronger generalization. Because optimal possibilities are explored in different ways, it is easier to adjust in the face of interference. Hence, for a $(s, a)$ pair sampled from $\mathcal{D}$, we obtain the target as

$$\hat{\mathcal{T}}_{\mathrm{DS}}^\pi Z_{\theta_m}(s, a) := R(s, a) + \gamma[\min_{j=1,\dots,M} Z_{\theta_j'}(s', a') - c\cdot\log\pi(a'|s')],$$

where the next $Z$-function takes the minimum value among the target $Z$-functions. The loss function is defined as follows,

$$\mathcal{L}_Z(\theta_m) = \mathcal{L}_p(Z_{\theta_m}, \hat{\mathcal{T}}_{\mathrm{DS}}^\pi Z_{\theta_m}).$$

Next, we introduce the smoothness regularizer term to the distribution action-value function that is designed to enhance the smoothness of the $Z$-function. Specifically, we minimize the difference between $Z_{\theta_m}(\hat{s}, a)$ and $Z_{\theta_m}(s, a)$, where $(\hat{s}, a)$ is a state-action pair with a perturbed state. Then we take an adversarial $\hat{s} \in \mathbb{B}_d(s, \epsilon)$ which maximizes $\mathcal{L}_p(Z_{\theta_m}(\hat{s}, a), Z_{\theta_m}(s, a))$. The final smooth term we introduced is shown below:

$$\mathcal{L}_{\mathrm{smooth}}(s, a; \theta_m) = \max_{\hat{s}\in\mathbb{B}_d(s,\epsilon)} \varrho \cdot \mathcal{L}_p(Z_{\theta_m}(\hat{s}, a), Z_{\theta_m}(s, a)), \tag{1}$$

where $\varrho \in [0, 1]$ is a factor that balances the learning of in-distribution and out-of-distribution values. Thus, for the selected $\hat{s} \in \mathbb{B}_d(s, \epsilon)$, we minimize Eq. (1) to get a smooth $Z$-function. Since the actions should be near the offline data and close to the behavior actions in the dataset, we do not consider OOD action for smoothing in this part.

Finally, we consider the following loss function to prevent overestimation of OOD actions.

$$\mathcal{L}_{\mathrm{OOD}}(\theta_m, \beta) = \beta \cdot \mathbb{E}_{U(\tau), \mathcal{D}(s,a)}[c_0(s, a) \cdot F_{Z_{\theta_m}(s,a)}^{-1}(\tau)],$$

for some state-action dependent scale facor $c_0$ and $U = \mathrm{Uniform}([0, 1])$.

Incorporating both the in-distribution target and OOD target, we conclude the loss function in ORDER as follows,

$$\min_{\theta_m} \mathbb{E}_{s,a,r,s'\sim\mathcal{D}}[\mathcal{L}_Z(\theta_m) + \alpha\mathcal{L}_{\mathrm{smooth}}(s, a; \theta_m) + \mathcal{L}_{\mathrm{OOD}}(\theta_m, \beta)]. \tag{2}$$

## 3.2 ROBUST POLICY

With the above smooth limits, we can learn a robust policy with fewer policy changes under perturbations. We choose a state $\hat{s} \in \mathbb{B}_d(s, \epsilon)$ as mentioned above which is maximizing $D_{\mathrm{J}}(\pi_\phi(\cdot|s)||\pi_\phi(\cdot|\hat{s}))$. Consequently, our loss function for policy is as follows:

$$\min_\phi[\mathbb{E}_{s,a,r,s'\sim\mathcal{D}}[-\min_{j=1,\dots,M} \Phi_g^j(s, a) + \alpha_1 \max_{\hat{s}\in\mathbb{B}_d(s,\epsilon)} D_{\mathrm{J}}(\pi_\phi(\cdot|s)||\pi_\phi(\cdot|\hat{s})) + \log\pi_\phi(a|s)]], \tag{3}$$

where the first term is designed to get a conservative policy by maximizing the minimum of the distributional functions ensemble and the last term is a regularization term.

## 3.3 IMPLEMENTATION DETAILS

In this subsection, we integrate the distributional evaluation and policy improvement algorithms introduced in Section 3.1 and Section 3.2 into the actor-critic framework. With the loss function introduced in Eq. (2) and Theorem 1, it is natural to get the following iterative formula for $Z^\pi$ by starting from an arbitrary $\tilde{Z}^0$,

$$\tilde{Z}^{k+1} = \arg\min_Z\{\mathcal{L}_p(Z(s,a), \hat{\mathcal{T}}_{DS}^\pi \hat{Z}^k(s,a)) + \max_{\hat{s}} \varrho \cdot \mathcal{L}_p(Z(\hat{s},a), Z(s,a))$$
$$+ \beta \cdot \mathbb{E}_{U(\tau),\mathcal{D}(s,a)}[c_0(s,a) \cdot F_{Z(s,a)}^{-1}(\tau)]\}. \tag{4}$$

Following Kumar et al. (2020), we suggest employing a min-max objective in which the inner loop selects the current policy to maximize the objective, while the outer loop is responsible for minimizing the objective with respect to this policy:

$$\tilde{Z}^{k+1} = \arg\min_Z \max_\mu \{\beta \cdot \mathbb{E}_{U(\tau)}[\mathbb{E}_{\mathcal{D}(s),\mu(\tilde{a}|s)} F_{Z(s,\tilde{a})}^{-1}(\tau) - \mathbb{E}_{\mathcal{D}(s,a)} F_{Z(s,a)}^{-1}]$$
$$+ \mathcal{L}_p(Z(s,a), \hat{\mathcal{T}}_{DS}^{\pi^k} \hat{Z}^k(s,a)) + \max_{\hat{s}} \varrho \cdot \mathcal{L}_p(Z(\hat{s},a), Z(s,a))\},$$

where $\mu$ is an actor policy. To establish a well-posed optimization problem, we introduce a regularization term in the original objective. Detailed analysis procedures are provided in Appendix A.1. The final optimization objective becomes

$$\tilde{Z}^{k+1} = \arg\min_Z\{\beta \cdot \mathbb{E}_{U(\tau)}[\mathbb{E}_{\mathcal{D}(s)} \log \sum_a \exp(F_{Z(s,a)}^{-1}(\tau)) - \mathbb{E}_{\mathcal{D}(s,a)} F_{Z(s,a)}^{-1}(\tau)]$$
$$+ \mathcal{L}_p(Z(s,a), \hat{\mathcal{T}}_{DS}^{\pi^k} \hat{Z}^k(s,a)) + \max_{\hat{s}} \varrho \cdot \mathcal{L}_p(Z(\hat{s},a), Z(s,a))\},$$

where $U = \text{Uniform}([0,1])$. To perform optimization with respect to the distribution $Z$, we express the quantile function using a deep neural network (DNN) $G_\theta(\tau; s, a) \approx F_{Z(s,a)}^{-1}(\tau)$. It has been demonstrated in (Koenker & Hallock, 2001) that $\mathbb{E}_{U(\tau)}\mathcal{L}_\kappa(\delta; \tau)$ is an unbiased estimator of the Wasserstein distance and can be optimized using stochastic gradient descent (SGD). Therefore, in order to calculate $\mathcal{L}_p(Z, Z') = W_p(Z, Z')^p$, we minimize the weighted pairwise Huber regression loss of various quantile fractions. For a sample $(s, a, r, s') \sim \mathcal{D}$ and $a' \sim \pi(\cdot|s')$ and random quantiles $\tau, \tau' \sim U$, distribution temporal differences (TD) error is defined as $\delta = r + \gamma G_{\theta'}(\tau'; s', a') - G_\theta(\tau; s, a)$. Then, $\tau$-Huber quantile regression loss (Huber, 1992) with threshold $\kappa$ is represented as

$$\mathcal{L}_\kappa(\delta; \tau) = \begin{cases} |\tau - \mathbf{1}\{\delta < 0\}| \cdot \delta^2/(2\kappa), & \text{if } |\delta| \leq \kappa, \\ |\tau - \mathbf{1}\{\delta < 0\}| \cdot (|\delta| - \kappa/2), & \text{otherwise.} \end{cases}$$

More details for the algorithm ORDER are presented in Algorithm 1.

## 3.4 THEORETICAL ANALYSIS

Before presenting our theorems, we first give some assumptions about the MDP and dataset. Next, we assume that the search space in Eq. (4) includes all possible functions.

**Assumption 1** *For all $s \in \mathcal{D}$ and $a \in \mathcal{A}$, $F_{Z^\pi(s,a)}$ is smooth. Furthermore, The search space of the minimum over $Z$ in Eq. (4) is overall smooth functions $F_{Z^\pi(s,a)}$ with support on $[V_{\min}, V_{\max}]$.*

The assumption is given to guarantee the boundness of the $p$-th moments of $Z^\pi(s,a)$ as well as $Z^\pi \in \mathcal{Z}$. Meanwhile, it is necessary for us to analyze and characterize the solution $\tilde{Z}^{k+1}$ of the objective of Eq. (4). Next, we assume that a stronger condition is needed.

**Assumption 2** *For all $s \in \mathcal{S}$ and $a \in \mathcal{A}$, there exists $\zeta \in \mathbb{R}_{>0}$, such that $F_{Z^\pi(s,a)}$ is $\zeta$-strongly monotone, i.e., $F'_{Z^\pi(s,a)}(x) \geq \zeta$.*

This assumption is only designed to ensure the convergence of $F_{Z^\pi(s,a)}^{-1}(x)$ in our theoretical analysis. Next, we assume that the infinite norm of two quantiles with perturbation is restricted by a fixed constant.

**Assumption 3** *For all $s \in \mathcal{D}$, $a \in \mathcal{A}$ and the selected $\hat{s} \in \mathbb{B}_d(s, \epsilon)$, we assume that $||F^{-1}_{Z(\hat{s},a)} - F^{-1}_{Z(s,a)}||_\infty \leq \sigma$, $\sigma$ is constant.*

In the above assumption, $|| \cdot ||_\infty$ represents the infinite norm. Since perturbed observation $\hat{s}$ is randomly sampling from an $\ell_\infty$ ball of norm $\epsilon$[1], the assumption is reasonable.

Finally, we assume $p > 1$. Therefore, we derive the following lemma which characterizes the conservative distribution soft evaluation iterates $Z(s, a)$ with the distributional soft Bellman operator.

**Lemma 1** *Suppose Assumptions 1-3 hold. For all $s \in \mathcal{D}$, $a \in \mathcal{A}$, $k \in \mathbb{N}$, and $\tau \in [0, 1]$, we have $F^{-1}_{Z(s,a)}(\tau) = F^{-1}_{\hat{\mathcal{T}}^\pi_{\mathrm{DS}}\hat{Z}^k(s,a)}(\tau) - c(s, a)$, where $c(s, a) = |\beta p^{-1} c_0(s, a) \pm \sigma^{p-1}|^{\frac{1}{p-1}} \cdot \mathrm{sign}(c_0(s, a))$.*

For detailed proof, please refer to Appendix B.1. Briefly, it is according to the result of a simple variational skill to handle that $F$ is a function, and setting the derivative of Eq. (4) equal to zero.

Next, we define the conservative soft distributional evaluation operator $\tilde{\mathcal{T}}^\pi = \mathcal{O}_c \hat{\mathcal{T}}^\pi_{\mathrm{DS}}$ by compositing $\hat{\mathcal{T}}^\pi_{\mathrm{DS}}$ and the shift operator $\mathcal{O}_c : \mathcal{Z} \to \mathcal{Z}$, which is defined by $F^{-1}_{\mathcal{O}_c Z(s,a)}(\tau) = F^{-1}_{Z(s,a)}(\tau) - c(s, a)$. Consequently, it is following that $\tilde{Z}^{k+1} = \tilde{\mathcal{T}}^\pi \tilde{Z}^k$. Next, we exhibit that $\tilde{\mathcal{T}}$ is contractive in $\bar{d}_p$ and $\tilde{Z}^\pi$ is the fixed point of $\tilde{\mathcal{T}}^\pi$.

**Theorem 1** *Under the same assumptions as in Lemma 1. $\tilde{\mathcal{T}}^\pi_{\mathrm{DS}}$ is a $\gamma$-contraction in $\bar{d}_p$, so $\tilde{Z}^k$ converges to a unique fixed point $\tilde{Z}^\pi$.*

For detailed proof, please refer to Appendix B.2. It takes advantage of the fact that $\hat{\mathcal{T}}^\pi$ is a a $\gamma$-contraction in $\bar{d}_p$ Bellemare et al. (2017); Dabney et al. (2018b) and the famous Banach fixed point theorem.

Now, the main theorem we came up with shows that the conservative distributional soft evaluation obtains a conservative quantile estimate of the true quantile at all quantiles $\tau$.

**Theorem 2** *Under the same assumptions as in Theorem 1. For any $\delta \in \mathbb{R}_{>0}, c_0(s, a) > 0$, with probability at least $1 - \delta$, we have*

$$F^{-1}_{Z^\pi(s,a)}(\tau) \geq F^{-1}_{\tilde{Z}^\pi(s,a)}(\tau) + (1-\gamma)^{-1} \min_{s',a'} \left\{ c(s', a') - \frac{1}{\zeta} \sqrt{\frac{5|\mathcal{S}|}{n(s', a')} \log \frac{4|\mathcal{S}||\mathcal{A}|}{\delta}} \right\},$$

*for all $s \in \mathcal{D}, a \in \mathcal{A}$, and $\tau \in [0, 1]$. Furthermore, for $\beta \geq \max_{s,a} \left\{ \frac{p(\Delta(s,a)^{p-1} + \sigma^{p-1})}{c_0(s,a)} \right\}$, we have $F^{-1}_{Z^\pi(s,a)}(\tau) \geq F^{-1}_{\tilde{Z}^\pi(s,a)}(\tau)$.*

For detailed proof, please refer to Appendix B.3. As the theorem shows, the above inequality indicates that the quantile estimates obtained by $\tilde{\mathcal{T}}^\pi$ are a lower bound of the true quantiles. Furthermore, we give a sufficient condition to show that the result given in Theorem 2 is not a vacuous conclusion. Therefore, Theorem 2 theoretically illustrates that ORDER does not exacerbate the distribution shift problem, and the mitigation of the distribution shift problem will be demonstrated in the experimental section.

The performance of many RL algorithms will exhibit different behaviors under different distorted expectations. Consequently, we can acquire the same conservative estimates of these objectives, which is a kind of generalization of Theorem 2.

**Corollary 1** *For any $\delta \in \mathbb{R}_{>0}$, $c_0(s, a) > 0$, sufficiently large $\beta$ and $g(\tau)$, with probability at least $1 - \delta$, for all $s \in \mathcal{D}, a \in \mathcal{A}$, we have $\Phi_g(Z^\pi(s, a)) \geq \Phi_g(\tilde{Z}^\pi(s, a))$.*

In particular, $Q^\pi(s, a) \geq \tilde{Q}^\pi(s, a)$ is obtained if we take $g = \mathrm{Uniform}([0, 1])$. By choosing different risk measure functions, we can apply this conclusion to any risk-sensitive offline RL.

---

[1] $\epsilon$ is usually a very small number.

## 4 RELATED WORKS

Offline RL (Lange et al., 2012; Wu et al., 2019; Kidambi et al., 2020; Prudencio et al., 2023) learns a policy from previously collected static datasets. As a subfield of RL (Sutton & Barto, 2018; Puterman, 2014), it has achieved significant accomplishments in practice (Singh et al., 2022; Diehl et al., 2023; Zhang et al., 2023; Singla et al., 2021). However, the main two challenges of offline RL are the distribution shift problem and robustness (Prudencio et al., 2023; Agarwal et al., 2020; Bai et al., 2022a), which require various techniques to improve the stability and performance of learned policies.

### 4.1 DISTRIBUTION SHIFT

The cause of the distribution shift problem is that the distribution of collected offline training data differs from the distribution of data in practice. BCQ (Fujimoto et al., 2019) addresses this problem through policy regularization techniques, which formulates the policy as an adaptable deviation constrained by maximum values (Sohn et al., 2015). One solution to alleviate the distribution shift problem as mentioned in BEAR (Kumar et al., 2019) is incorporating a weighted behavior-cloning loss achieved by minimizing maximum mean discrepancy (MMD) into the policy improvement step. Though learning a conservative $Q$-function caused by distribution shift, CQL (Kumar et al., 2020) solves the overestimation of value functions, which theoretically proves that a lower bound of the true value is obtained. With the introduction of distributional reinforcement learning into offline RL, CODAC (Ma et al., 2021) learns a conservative return distribution by punishing the predicted quantiles returned for the OOD actions. From another perspective, continuous quantiles are used in MQN-CQR (Bai et al., 2022b) to learn the quantile of return distribution with non-crossing guarantees. ORDER builds on these approaches, but considers the entropy regularizer of quantile function instead of an unchangeable constant for ensuring sufficient exploration and may relieve the training imbalance.

### 4.2 ROBUSTNESS ISSUES

Owning to the distribution shift issues, current offline RL algorithms tend to prioritize caution in their approach to value estimation and action selection. Nevertheless, this selection can compromise the robustness of learned policies, making them highly sensitive to even minor perturbations in observations. As a groundbreaking piece of work, $\text{SR}^2\text{L}$ (Shen et al., 2020) achieves a more robust policy by introducing a smoothness regularizer into both the value function and the policy. A robust $Q$-learning algorithm proposed in REM (Agarwal et al., 2020) is presented that integrates multiple $Q$-value networks in a random convex combination of multiple $Q$-value estimates, ensuring that the final $Q$-value estimate remains robust. With arbitrarily large state spaces, RFQI (Panaganti et al., 2022) learns the optimal policy by employing function approximation using only an offline dataset and addresses robust offline reinforcement learning problems. In ORDER, we import the smoothing regularizer to the distribution functions and policies instead of simple action-value functions.

Our method is related to the previous offline RL algorithms based on constraining the learned value function (Kostrikov et al., 2021; Kumar et al., 2020). What sets our method apart is that it can better capture uncertain information about OOD actions and learn more robust policies with the introduction of the smoothing technique into distributional reinforcement learning. In addition, we enhance the network by using the entropy regularizer of quantile networks and increasing the number of network ensembles, which also improves robustness.

## 5 EXPERIMENTS

In the sequel, we first compare ORDER against some offline RL algorithms. Then how different risk measure functions impact our proposed algorithm is investigated in Section 5.2. Besides, Section 5.3 shows the ensemble size of the quantile network in ORDER. And our approach significantly exceeds the baseline algorithm in most tasks.

We evaluate our experiments on the D4RL benchmark (Fu et al., 2020) with various continuous control tasks and datasets. Specifically, we employ three environments (HalfCheetah, Hopper and Walker2d) and four dataset types (random, medium, medium-replay, and medium-expert). The

random or medium dataset is generated by a single random or medium policy. The medium-replay dataset contains experiences collected in training a medium-level policy, and the medium-expert dataset is a mixture of medium and expert datasets.

Table 1: Normalized average returns on D4RL benchmark, averaged over five random seeds. "r","m","m-r" and "m-e" indicate the abbreviations of random, medium, medium-replay and medium-expert, respectively. All methods are run for 1M gradient steps.

| Datasets | BEAR | CQL | CODAC | RORL (Reproduced) | MQN-CQR | ORDER |
|---|---|---|---|---|---|---|
| hopper-r | 3.9±2.3 | 7.9±0.4 | 11.0±0.4 | **22.7±8.4** | 13.2±0.6 | **24.8 ± 7.8** |
| hopper-m | 51.8±4.0 | 53.0±28.5 | 70.8±11.4 | **104.8±0.3** | 94.7±13.2 | 101.5±0.2 |
| hopper-m-r | 52.2±19.3 | 88.7±12.9 | 100.2±1.0 | 102.3±0.5 | 95.6±18.5 | **106.4±0.1** |
| hopper-m-e | 50.6±25.3 | 105.6±12.9 | **112.0±1.7** | 112.8±0.2 | 113.0±0.5 | **114.6±3.3** |
| walker2d-r | 12.8±10.2 | 5.1±1.3 | 18.7±4.5 | 21.5±0.2 | 22.6±6.1 | **28.4± 6.2** |
| walker2d-m | -0.2±0.1 | 73.3±17.7 | 82.0±0.5 | **103.2±1.7** | 80.0±0.5 | 86.0± 0.2 |
| walker2d-m-r | 7.0±7.8 | 81.8±2.7 | 33.2±17.6 | **90.1±0.6** | 52.3±16.7 | **87.9±4.8** |
| walker2d-m-e | 22.1±44.9 | 107.9±1.6 | 106.0±4.6 | **120.3±1.8** | 112.1±8.9 | 115.1±1.2 |
| halfCheetah-r | 2.3±0.0 | 17.5±1.5 | **34.6±1.3** | 28.2±0.7 | 32.6±2.9 | 31.5±1.0 |
| halfCheetah-m | 43.0±0.2 | 47.0±0.5 | 46.3±1.0 | **64.7±1.1** | 45.1±1.5 | **63.7±0.4** |
| halfCheetah-m-r | 36.3±3.1 | 45.5±0.7 | 44.0±0.8 | **61.1±0.7** | 45.3±7.9 | 57.4±1.7 |
| halfCheetah-m-e | 46.0±4.7 | 75.6±25.7 | 70.4±19.4 | **108.2±0.8** | 71.1±4.9 | 93.2±1.1 |

## 5.1 COMPARISON WITH OFFLINE RL ALGORITHMS

In all the aforementioned datasets, we compare our method against several previous popular offline RL algorithms, including (i) bootstrapping error accumulation reduction (BEAR) (Kumar et al., 2019), (ii) conservative q-learning (CQL) (Kumar et al., 2020), (iii) CODAC (Ma et al., 2021), (iv) robust offline reinforcement learning (RORL) (Yang et al., 2022), (v) monotonic quantile network with conservative quantile regression (MQN-CQR) (Bai et al., 2022b). The results of BEAR and CQL are directly taken from (Fu et al., 2020). For CODAC and MQN-CQR, their results are taken from the original paper. Since the RORL paper does not report scores with five random seeds, we run the RORL using the official code base. The neural network architecture we use is given in Appendix C.2. Table 4 and Table 5 list the hyperparameters of ORDER in different datasets. Without loss of generality, we employ the neutral risk measure in this subsection.

The performance of all these algorithms is exhibited in Table 1, which reports the average normalized scores along with their corresponding standard deviations. We observe that ORDER outperforms BEAR in all tasks and surpasses the performance of the CQL algorithm. Significantly, our algorithm surpasses the performance of current distributed offline RL methods (see CODAC and MQN-CQR in Table 1), which is attributed to the assurance of robustness in ORDER. Meanwhile, ORDER competes favorably with the current state-of-the-art algorithms, owing to the safety guaranteed by distributional RL.

## 5.2 POLICY TRAINING UNDER RISK MEASURES FUNCTION

In this subsection, we investigate how the risk measure functions affect the performance of ORDER. We compare three risk-averse learned policies (Ma et al., 2020) in distribution RL with the risk-neutral measure function. Specifically, for different $g(\tau)$, three ways of distorted expectation are considered,

- CPW: $g(\tau) = \tau^\lambda/(\tau^\lambda + (1-\tau)^\lambda)^{1/\lambda}$, and $\lambda$ is set as 0.71.
- Wang: $g(\tau) = F_\mathcal{N}(F_\mathcal{N}^{-1}(\tau) + \lambda)$, where $\lambda$ is set as 0.25 and $F_\mathcal{N}$ is the standard Gaussian CDF.
- CVaR: $g(\tau)=\min\{\tau/\lambda, 1\}$, and $\lambda$ is set as 0.25.

Besides, we evaluate three risk-seeking learned policies.

- The first risk-seeking method is mean-variance and $\lambda$ is set as -0.1.
- The second risk-seeking method is Var, and $\lambda$ is set as 0.75.
- The third risk-seeking method is Wang, and $\lambda$ is set as -0.75.

Table 2: Performance of ORDER under various risk-averse methods in hopper-medium-replay-v2. Each method is run with five random seeds.

| Risk measure | Neutral | CPW(0.71) | CVaR(0.25) | Wang(0.25) |
|---|---|---|---|---|
| Performance of ORDER | 106.4±0.1 | 105.3±0.4 | 106.2±0.4 | 106.0±1.5 |

Table 3: Performance of ORDER under various risk-seeking methods in hopper-medium-replay-v2. Each method is run with five random seeds.

| Risk measure | Neutral | Mean-Std(-0.1) | VaR(0.75) | Wang(-0.75) |
|---|---|---|---|---|
| Performance of ORDER | 106.4 ±0.1 | 107.5±0.3 | 106.5±0.6 | 106.4±0.2 |

The results are shown in Table 2 and Table 3, indicating that there is little difference between risk-averse methods and risk-seeking learned policies. This suggests that risk measure functions within the ORDER framework are not highly sensitive. At this point, we also empirically demonstrate the robustness of our approach.

## 5.3 ABLATIONS ON BENCHMARK RESULTS

Without loss of generality, we choose the hopper-medium-replay-v2 dataset as an example to conduct the ablation study in this subsection. The performance of our ORDER algorithm under different $M$s is visualized in Fig. 2. We observe with the increase of $M$, the effect has a significant improvement in both computation efficiency and stability; as shown in the yellow and purple lines. However, $M$ should not be too large, which is presumably attributed to the overfitting problem (see the blue line, the normalized score value fluctuates significantly around the training epoch of 700). In conclusion, $M$ is set as four to balance between robustness enhancement and computational efficiency improvement.

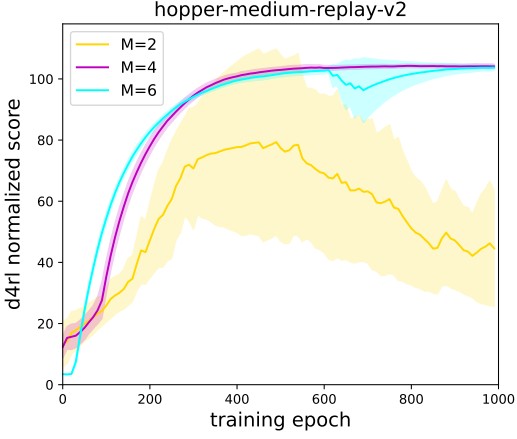

Figure 2: The normalized score under different ensemble sizes. Each method is run with five random seeds.

## 6 CONCLUSION

In this work, we introduce Offline Robustness of Distributional actor-critic Ensemble Reinforcement Learning (ORDER) to balance the conservatism and robustness in the offline setting. To achieve robustness, we first take into account the entropy regularizer that helps fully explore the dataset and alleviates training imbalance issues. Moreover, we consider the ensemble of multiple quantile networks to enhance robustness. Furthermore, a smoothing technique is introduced to the policies and the distributional functions for the perturbed states. In addition, we theoretically prove that ORDER converges to a conservative lower bound, which also shows that we improve the robustness without exacerbating the OOD problem. Finally, ORDER shows its advantage against the existing distributional offline RL methods in the D4RL benchmark. We also validate the effectiveness of ORDER through ablation studies.

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

# A    ALGORITHM AND IMPLEMENTATION DETAILS

In this section, we provide a comprehensive account of our practical implementation of ORDER, offering a detailed explanation of the process.

## A.1    ORDER OBJECTIVE

To establish a well-defined optimization problem, we introduce a regularization term denoted as $\mathcal{R}(\mu)$ in the original objective:

$$\hat{Z}^{k+1} = \arg\min_{Z} \max_{\mu} \{\beta \cdot \mathbb{E}_{U(\tau)}[\mathbb{E}_{\mathcal{D}(s),\mu(\tilde{a}|s)} F_{Z(s,\tilde{a})}^{-1}(\tau) - \mathbb{E}_{\mathcal{D}(s,a)} F_{Z(s,a)}^{-1}(\tau)]$$
$$+ \mathcal{L}_p(Z, \hat{\mathcal{T}}^{\pi^k} \hat{Z}^k) + \max_{\hat{s}} \varrho \cdot \mathcal{L}_p(Z(\hat{s},a), Z(s,a))\} + \mathcal{R}(\mu).$$

Let $c_0(s,a) = \frac{\mu(a|s)-\hat{\pi}(a|s)}{\hat{\pi}(a|s)}$ and $\mathcal{R}(\mu)$ to be the entropy $\mathcal{H}(\mu)$, then $\mu(a|s) \propto \exp\left(Q(s,a)\right) = \exp\left(\int_0^1 F_{Z(s,a)}^{-1}(\tau)d\tau\right)$ is the solution to the inner-maximization. Substituting this selection into the previously mentioned regularized objective function gives

$$\hat{Z}^{k+1} = \arg\min_{Z} \{\beta \cdot \mathbb{E}_{U(\tau)}[\mathbb{E}_{\mathcal{D}(s)} \log \sum_a \exp(F_{Z(s,a)}^{-1}(\tau)) - \mathbb{E}_{\mathcal{D}(s,a)} F_{Z(s,a)}^{-1}(\tau)]$$
$$+ \mathcal{L}_p(Z, \hat{\mathcal{T}}^{\pi^k} \hat{Z}^k) + \max_{\hat{s}} \varrho \cdot \mathcal{L}_p(Z(\hat{s},a), Z(s,a))\}.$$

As demonstrated in (Ma et al., 2021), we also introduce a parameter $\zeta \in \mathbb{R}_{>0}$ to threshold the quantile value difference between $\mu$ and $\hat{\pi}$, and give this difference a weight $\xi \in \mathbb{R}_{>0}$. Then we get a trainable expression of $\beta$ through the process of dual gradient descent:

$$\min_{Z} \max_{\beta \geq 0} \{\beta \cdot \mathbb{E}_{U(\tau)}[\xi \cdot [\mathbb{E}_{\mathcal{D}(s)} \log \sum_a \exp(F_{Z(s,a)}^{-1}(\tau)) - \mathbb{E}_{\mathcal{D}(s,a)} F_{Z(s,a)}^{-1}(\tau)] - \zeta]$$
$$+ \mathcal{L}_p(Z, \hat{\mathcal{T}}^{\pi^k} \hat{Z}^k) + \max_{\hat{s}} \varrho \cdot \mathcal{L}_p(Z(\hat{s},a), Z(s,a))\}.$$

Since all our experiments take place in continuous-control domains, it is not feasible to list all possible actions $a$s and directly compute $\log \sum_a \exp(F_{Z(s,a)}^{-1}(\tau))$. In our implementation, we employ the importance sampling approximation method described in (Kumar et al., 2020), and obtain

$$\log \sum_a \exp(F_{Z(s,a)}^{-1}(\tau)) \approx \log\left(\frac{1}{2C} \sum_{a_i \sim U(\mathcal{A})}^{N} \left[\frac{\exp(F_{Z(s,a)}^{-1}(\tau))}{U(\mathcal{A})}\right] + \frac{1}{2C} \sum_{a_i \sim \pi(a|s)}^{N} \left[\frac{\exp(F_{Z(s,a)}^{-1}(\tau))}{\pi(a_i|s)}\right]\right),$$
(5)

where $U(\mathcal{A}) = \text{Uniform}(\mathcal{A})$ represents a uniform distribution of actions. Algorithm 1 summarizes a single step of the actor and critic updates used by ORDER.

---

**Algorithm 1** ORDER

---

1: **Hyperparameters:** Number of generated quantiles $N$, number of quantile networks for value functions $M$, Huber loss threshold $\kappa$, discount rate $\gamma$, learning rates $\eta_{\text{actor}}, \eta_Z, \eta_\beta$, tuning parameter $\alpha, \alpha_1$, OOD penalty scale $\xi$, OOD penalty threshold $\zeta$
2: **Parameters:** Critic parameters $\theta$, Actor parameters $\phi$, Penalty $\beta$
3: # Compute distributional TD loss
4: Get the next action using current policy $a' \sim \pi(\cdot|s'; \phi)$
5: **for** $i = 1$ **to** $N$ **do** (For the $m$-th quantile network)
6:     **for** $j = 1$ **to** $N$ **do**
7:         $\delta^m_{\tau_i, \tau'_j} = r + \gamma F^{-1}_{Z(s',a'),\theta'_m}(\tau'_j) - F^{-1}_{Z(s,a),\theta_m}(\tau_i)$
8:     **end for**
9: **end for**
10: Computer $\mathcal{L}_{\text{critic}}(\theta_m) = N^{-2} \sum_{i=1}^{N} \sum_{j=1}^{N} \mathcal{L}_\kappa(\delta^m_{\tau_i, \tau'_j}; \tau_i)$
11: # Compute OOD penalty
12: Sample $i \sim U(\{1, \ldots, N\})$ and use quantile $\tau_i$
13: Estimate $\log \sum_a \exp(F^{-1}_{Z(s,a),\theta_m}(\tau_i))$ according to Eq. (5)
14: Compute

$$\mathcal{L}_{\text{OOD}}(\theta_m, \beta) = \beta \cdot \left( \xi \cdot \left( \log \sum_a \exp(F^{-1}_{Z(s,a),\theta_m}(\tau_i)) - N^{-1} \sum_{j=1}^{N} F^{-1}_{Z(s,a),\theta_m}(\tau_j) \right) - \zeta \right).$$

15: # Update quantile network
16: Use Eq. (1) to add perturbations to the state to obtain $\hat{s}$.
17: Train $\theta$ using Eq. (2) by SGD
18: Update $\theta \leftarrow \theta - \eta_Z \nabla(\mathcal{L}_Z(\theta_m) + \alpha \mathcal{L}_{\text{smooth}}(s, a; \theta_m) + \mathcal{L}_{\text{OOD}}(\theta_m, \beta))$
19: # Update policy network with $\Phi_j$ objective
20: Get new actions with re-parameterized samples $\tilde{a} \sim \pi(\cdot|s; \phi)$
21: Computer $\Phi_g(s, \tilde{a})$ using $F^{-1}_{Z(s,\tilde{a}),\theta}(\tau_i), i = 1, \ldots, N$
22: $\mathcal{L}_{\text{actor}}(\phi) = \log(\pi(\tilde{a}|s; \phi)) + \alpha_1 \max_{\hat{s}} D_J(\pi_\phi(\cdot|s)||\pi_\phi(\cdot|\hat{s})) - \min_{j=1,\ldots,M} \Phi^j_g(s, \tilde{a})$
23: Update $\phi \leftarrow \phi + \eta_{\text{actor}} \nabla \mathcal{L}_{\text{actor}}(\phi)$

---

# B    PROOFS

## B.1    PROOF OF LEMMA 1

*Proof.* By the definition of $p$-Wasserstein distance, we can re-write Eq. (4) as

$$\mathbb{E}_{\mathcal{D}(s,a)} \int_0^1 |F^{-1}_{Z(s,a)}(\tau) - F^{-1}_{\hat{\mathcal{T}}^\pi_{DS} \hat{Z}^k(s,a)}(\tau)|^p d\tau + \beta \cdot \mathbb{E}_{U(\tau), \mathcal{D}(s,a)}[c_0(s,a) \cdot F^{-1}_{Z(s,a)}(\tau)]$$

$$+ \mathbb{E}_{\mathcal{D}(s,a), \hat{s} \in \mathbb{B}_d(s,\epsilon)} \int_0^1 |F^{-1}_{Z(\hat{s},a)}(\tau) - F^{-1}_{Z(s,a)}(\tau)|^p d\tau$$

$$= \int_0^1 \mathbb{E}_{\mathcal{D}(s,a)}[|F^{-1}_{Z(s,a)}(\tau) - F^{-1}_{\hat{\mathcal{T}}^\pi_{DS} \hat{Z}^k(s,a)}(\tau)|^p + \beta \cdot c_0(s,a) \cdot F^{-1}_{Z(s,a)}(\tau)$$

$$+ |F^{-1}_{Z(\hat{s},a)}(\tau) - F^{-1}_{Z(s,a)}(\tau)|^p] d\tau.$$

For arbitrary smooth functions $\phi_{s,a}$ with compact support $[V_{\min}, V_{\max}]$, we consider a perturbation $G^\varepsilon_{s,a}(\tau) = F^{-1}_{Z(s,a)}(\tau) + \varepsilon \cdot \phi_{s,a}(\tau)$ of $F^{-1}_{Z(s,a)}(\tau)$, then the above formula can be written as

$$\int_0^1 \mathbb{E}_{\mathcal{D}(s,a)}[|G^\varepsilon_{s,a}(\tau) - F^{-1}_{\hat{\mathcal{T}}^\pi_{DS} \hat{Z}^k(s,a)}(\tau)|^p + \beta \cdot c_0(s,a) \cdot G^\varepsilon_{s,a}(\tau) + |G^\varepsilon_{\hat{s},a}(\tau) - G^\varepsilon_{s,a}(\tau)|^p] d\tau.$$

Then the following equation is obtained considering the derivative of $\varepsilon$ at $\varepsilon = 0$:

$$\frac{d}{d\varepsilon} \int_0^1 \mathbb{E}_{\mathcal{D}(s,a)}[|G_{s,a}^\varepsilon(\tau) - F_{\hat{\mathcal{T}}_{DS}^\pi \hat{Z}^k(s,a)}^{-1}(\tau)|^p + \beta c_0(s,a) \cdot G_{s,a}^\varepsilon(\tau) + |G_{\hat{s},a}^\varepsilon(\tau) - G_{s,a}^\varepsilon(\tau)|^p]d\tau|_{\varepsilon=0}$$

$$= \mathbb{E}_{\mathcal{D}(s,a)} \int_0^1 [p|F_{Z(s,a)}^{-1}(\tau) - F_{\hat{\mathcal{T}}_{DS}^\pi \hat{Z}^k(s,a)}^{-1}(\tau)|^{p-1} \text{sign}(F_{Z(s,a)}^{-1}(\tau) - F_{\hat{\mathcal{T}}_{DS}^\pi \hat{Z}^k(s,a)}^{-1}(\tau))$$

$$+ \beta c_0(s,a) + p|F_{Z(\hat{s},a)}^{-1}(\tau) - F_{Z(s,a)}^{-1}(\tau)|^{p-1}\text{sign}(F_{Z(\hat{s},a)}^{-1}(\tau) - F_{Z(s,a)}^{-1}(\tau))]\phi_{s,a}(\tau)d\tau.$$

Owning to some perturbation $G_{s,a}^\varepsilon$ will cause the objective value to decrease, so the above equation must be equal to 0 for $F_{Z(s,a)}^{-1}$. If $\phi(s,a)$ does not equal zero for each $s, a$, since $\phi(s,a)$ is arbitrary, it will also cause the above equation to be not equal to 0. Therefore, we obtain

$$\int_0^1 [p|F_{Z(s,a)}^{-1}(\tau) - F_{\hat{\mathcal{T}}_{DS}^\pi \hat{Z}^k(s,a)}^{-1}(\tau)|^{p-1} \text{sign}(F_{Z(s,a)}^{-1}(\tau) - F_{\hat{\mathcal{T}}_{DS}^\pi \hat{Z}^k(s,a)}^{-1}(\tau))$$

$$+ \beta c_0(s,a) + p|F_{Z(\hat{s},a)}^{-1}(\tau) - F_{Z(s,a)}^{-1}(\tau)|^{p-1}\text{sign}(F_{Z(\hat{s},a)}^{-1}(\tau) - F_{Z(s,a)}^{-1}(\tau))]\phi_{s,a}(\tau)d\tau = 0.$$

for all $s, a$. According to the above term is zero for all $\phi(s,a)$, we have

$$p|F_{Z(s,a)}^{-1}(s,a)(\tau) - F_{\hat{\mathcal{T}}_{DS}^\pi \hat{Z}^k(s,a)}^{-1}(\tau)|^{p-1}\text{sign}(F_{Z(s,a)}^{-1}(\tau) - F_{\hat{\mathcal{T}}_{DS}^\pi \hat{Z}^k(s,a)}^{-1}(\tau))$$

$$+ \beta c_0(s,a) + p|F_{Z(\hat{s},a)}^{-1}(\tau) - F_{Z(s,a)}^{-1}(\tau)|^{p-1}\text{sign}(F_{Z(\hat{s},a)}^{-1}(\tau) - F_{Z(s,a)}^{-1}(\tau)) = 0.$$

According to Assumption 3, the above equation can be converted to

$$p|F_{Z(s,a)}^{-1}(\tau) - F_{\hat{\mathcal{T}}_{DS}^\pi \hat{Z}^k(s,a)}^{-1}(\tau)|^{p-1}\text{sign}(F_{Z(s,a)}^{-1}(\tau) - F_{\hat{\mathcal{T}}_{DS}^\pi \hat{Z}^k(s,a)}^{-1}(\tau)) = \beta c_0(s,a) \pm \sigma^{p-1},$$

which holds if and only if

$$F_{Z(s,a)}^{-1}(\tau) = F_{\hat{\mathcal{T}}_{DS}^\pi \hat{Z}^k(s,a)}^{-1}(\tau) - c(s,a),$$

where $c(s,a) = |\beta p^{-1} c_0(s,a) \pm \sigma^{p-1}|^{\frac{1}{p-1}} \cdot \text{sign}(c_0(s,a))$. $\qquad\square$

### B.2 PROOF OF THEOREM 1

*Proof.* Let $Z_1, Z_2 \in \mathcal{Z}$ denote two action-value distributions. For all $(s,a) \in \mathcal{S} \times \mathcal{A}$, we have

$$W_p(\hat{\mathcal{T}}_{DS}^\pi Z_1(s,a), \hat{\mathcal{T}}_{DS}^\pi Z_2(s,a))$$

$$= W_p(R(s,a) + \gamma[\min_{j=1,\dots,M} Z_{1,\theta_j'}(s',a') - c \cdot \log \pi(a'|s')],$$

$$R(s,a) + \gamma[\min_{j=1,\dots,M} Z_{2,\theta_j'}(s',a') - c \cdot \log \pi(a'|s')])|s' \sim \mathbb{P}(\cdot|s,a), a' \sim \pi(\cdot|s')$$

$$\leq \gamma W_p(Z_1(s',a'), Z_2(s',a'))$$

$$\leq \gamma \sup_{s',a'} W_p(Z_1(s',a'), Z_2(s',a')).$$

By the definition of $\bar{d}_p$, we have

$$\bar{d}_p(\hat{\mathcal{T}}_{DS}^\pi Z_1, \hat{\mathcal{T}}_{DS}^\pi Z_2) = \sup_{s,a} W_p(\hat{\mathcal{T}}_{DS}^\pi Z_1(s,a), \hat{\mathcal{T}}_{DS}^\pi Z_2(s,a))$$

$$\leq \gamma \sup_{s',a'} W_p(Z_1(s',a'), Z_2(s',a'))$$

$$= \gamma \bar{d}_p(Z_1, Z_2).$$

Therefore, $\hat{\mathcal{T}}_{DS}^\pi$ is a $\gamma$-contraction in $\bar{d}_p$. Since $\mathcal{O}_c$ is a non-expansion is $\bar{d}_p$, then $\tilde{\mathcal{T}}_{DS}^\pi$ is a $\gamma$-contraction in $\bar{d}_p$. By Banach fixed point theorem, $\tilde{Z}^k$ converges to a unique fixed point $\tilde{Z}^\pi$. $\qquad\square$

### B.3 PROOF OF THEOREM 2

**Lemma 2** *For all $\delta \in \mathbb{R}_{>0}$, with probability at least $1 - \delta$, for any $Z \in \mathcal{Z}$ and $(s,a) \in \mathcal{D}$, we have*

$$||F_{\hat{\mathcal{T}}^\pi Z(s,a)} - F_{\mathcal{T}^\pi Z(s,a)}||_\infty \leq \sqrt{\frac{5|\mathcal{S}|}{n(s,a)} \log \frac{4|\mathcal{S}||\mathcal{A}|}{\delta}}, \tag{6}$$

*where $n(s,a)$ represents the number of occurrences of $(s,a)$ in $\mathcal{D}$.*

*Proof.* Applying the definition of distributional soft Bellman operator to the cumulative density function, we obtain that

$$
F_{\hat{\mathcal{T}}_{\mathrm{DS}}^\pi Z(s,a)}(x) - F_{\mathcal{T}_{\mathrm{DS}}^\pi Z(s,a)}(x)
$$
$$
= \sum_{s',a'} \hat{P}(s'|s,a)\pi(a'|s')F_{\gamma[Z(s',a')-c\cdot\log\pi(a'|s')]+\hat{R}(s,a)}(x)
$$
$$
- \sum_{s',a'} P(s'|s,a)\pi(a'|s')F_{\gamma[Z(s',a')-c\cdot\log\pi(a'|s')]+R(s,a)}(x).
$$

Adding and subtracting $\sum_{s',a'} \hat{P}(s'|s,a)\pi(a'|s')F_{\gamma[Z(s',a')-c\cdot\log\pi(a'|s')]+R(s,a)}(x)$ from this expression gives

$$
\sum_{s',a'} \hat{P}(s'|s,a)\pi(a'|s')(F_{\gamma[Z(s',a')-c\cdot\log\pi(a'|s')]+\hat{R}(s,a)}(x) - F_{\gamma[Z(s',a')-c\cdot\log\pi(a'|s')]+R(s,a)}(x))
$$
$$
+ \sum_{s',a'} (\hat{P}(s'|s,a) - P(s'|s,a))\pi(a'|s')F_{\gamma[Z(s',a')-c\cdot\log\pi(a'|s')]+R(s,a)}(x).
$$

We proceed by bounding the two terms in the summation. For the first term,

$$
F_{\gamma[Z(s',a')-c\cdot\log\pi(a'|s')]+\hat{R}(s,a)}(x) - F_{\gamma[Z(s',a')-c\cdot\log\pi(a'|s')]+R(s,a)}(x)
$$
$$
= \int \left[ F_{\hat{R}(s,a)}(r) - F_{R(s,a)}(r) \right] dF_{\gamma[Z(s',a')-c\cdot\log\pi(a'|s')]}(x-r)
$$
$$
\le \int |F_{\hat{R}(s,a)}(r) - F_{R(s,a)}(r)| dF_{\gamma[Z(s',a')-c\cdot\log\pi(a'|s')]}(x-r)
$$
$$
\le \sup_r |F_{\hat{R}(s,a)}(r) - F_{R(s,a)}(r)| \int dF_{\gamma[Z(s',a')-c\cdot\log\pi(a'|s')]}(x-r)
$$
$$
= ||F_{\hat{R}(s,a)}(r) - F_{R(s,a)}(r)||_\infty.
$$

Therefore,

$$
\sum_{s',a'} \hat{P}(s'|s,a)\pi(a'|s')(F_{\gamma[Z(s',a')-c\cdot\log\pi(a'|s')]+\hat{R}(s,a)}(x) - F_{\gamma[Z(s',a')-c\cdot\log\pi(a'|s')]+R(s,a)}(x))
$$
$$
\le \sum_{s',a'} \hat{P}(s'|s,a)\pi(a'|s')||F_{\hat{R}(s,a)}(r) - F_{R(s,a)}(r)||_\infty
$$
$$
= ||F_{\hat{R}(s,a)}(r) - F_{R(s,a)}(r)||_\infty.
$$

The following derivation process is similar to CODAC (Ma et al., 2021), so we can finally obtain the above conclusion. □

It has been proved in CODAC that if $||F - G||_\infty \le \epsilon$, then $||F^{-1} - G^{-1}||_\infty \le \epsilon/\zeta$, where $F$ and $G$ are two cumulative distribution functions (CDFs) with support $\chi$, and $F$ is $\zeta$-strongly monotone. Thus, according to Lemma 2, we have

**Lemma 3** *For any return distributional $Z$ with $\zeta$-strongly monotone CDF $F_{Z(s,a)}$ and any $\delta \in \mathbb{R}_{>0}$, with probability at least $1 - \delta$, for all $s \in \mathcal{D}$ and $a \in \mathcal{A}$, we have*

$$
||F^{-1}_{\hat{\mathcal{T}}_{\mathrm{DS}}^\pi Z(s,a)} - F^{-1}_{\mathcal{T}^\pi Z(s,a)}||_\infty \le \frac{1}{\zeta}\sqrt{\frac{5|\mathcal{S}|}{n(s,a)}\log\frac{4|\mathcal{S}||\mathcal{A}|}{\delta}}.
$$

Let $\Delta(s,a) = \frac{1}{\zeta}\sqrt{\frac{5|\mathcal{S}|}{n(s,a)}\log\frac{4|\mathcal{S}||\mathcal{A}|}{\delta}}$ and followed by Lemma 1

$$
F^{-1}_{\tilde{\mathcal{T}}^\pi Z^\pi(s,a)}(\tau) = F^{-1}_{\hat{\mathcal{T}}_{\mathrm{DS}}^\pi Z^\pi(s,a)}(\tau) - c(s,a)
$$
$$
\le F^{-1}_{\mathcal{T}^\pi Z^\pi(s,a)}(\tau) - c(s,a) + \Delta(s,a)
$$
$$
= F^{-1}_{Z^\pi(s,a)}(\tau) - c(s,a) + \Delta(s,a),
$$

the second step holds by Lemma 2 with probability at least $1 - \delta$. For any $h \in \mathbb{R}$, if $Z$ satisfies $F_{Z(s,a)}^{-1}(\tau) \geq F_{\mathcal{T}^\pi Z(s,a)}^{-1}(\tau) + h$ for all $s \in \mathcal{S}$ and $a \in \mathcal{A}$, then $F_{Z(s,a)}^{-1}(\tau) \geq F_{\mathcal{T}^\pi Z(s,a)}^{-1}(\tau) + (1 - \gamma)^{-1}h(\forall \tau \in [0,1])$. Then,

$$
\begin{aligned}
F_{Z^\pi(s,a)}^{-1}(\tau) &\geq F_{\tilde{\mathcal{T}}^\pi Z^\pi(s,a)}^{-1}(\tau) + c(s,a) - \Delta(s,a) \\
&\geq F_{\tilde{\mathcal{T}}^\pi Z^\pi(s,a)}^{-1}(\tau) + \min_{s,a}\{c(s,a) - \Delta(s,a)\} \\
&\geq F_{\tilde{Z}^\pi(s,a)}^{-1}(\tau) + (1 - \gamma)^{-1}\min_{s,a}\{c(s,a) - \Delta(s,a)\}.
\end{aligned} \tag{7}
$$

Notice that since for the last term in Eq. (7) to be positive, we need

$$
\beta p^{-1} c_0(s,a) + \sigma^{p-1} \geq \Delta(s,a)^{p-1} \quad (\forall s, a).
$$

Owning to we have assumed that $c_0(s,a) > 0$, then it can be equivalent to

$$
\beta \geq \max_{s,a}\{\frac{p(\Delta(s,a)^{p-1} + \sigma^{p-1})}{c_0(s,a)}\},
$$

we thus prove the conclusion of Theorem 2.

## C  EXPERIMENTAL SETTINGS AND IMPLEMENTATION DETAILS

### C.1  EXPERIMENTAL SETTINGS

Our experimental procedure largely adheres to (Fu et al., 2020), and the results of non-distributional methods are directly taken from (Fu et al., 2020). For all experiments, we run algorithms for 1000 epochs (1000 training steps each epoch, i.e., 1M gradient steps in total). Then we evaluate them using 10 test episodes in the original environment, which all last 1000 steps long. All benchmark results are averaged over 5 random seeds. The reported results are normalized to D4RL scores that measure how the performance compared with expert score and random score: normalized score = $100 \times \frac{\text{score} - \text{random score}}{\text{expert score} - \text{random score}}$.

### C.2  NEURAL NETWORK ARCHITECTURE

The same network architecture of ORDER is used in all experiments. For policy, we use a two-layer fully connected neural network with 256 hidden neurons and ReLU activations. For the quantile network, we adopt an ensemble structure. Specifically, the quantile function is the Hadamard product of state-action feature $\psi(s,a)$ and quantile embedding $\varphi(\tau)$, i.e., $F_{Z(s,a)}^{-1}(\tau) = \psi(s,a) \odot \varphi(\tau)$. Embedding formula of $\varphi(\tau)$ represent as $\varphi_j(\tau) := h(\sum_{i=1}^{n} \cos(i\pi\tau)w_{ij} + b_j)$(Dabney et al., 2018a), where $w_{ij}, b_j$ are weights of the neural network $\varphi$, and $h$ is the sigmoid function. $\varphi(\tau)$ consists of a one-layer 64-unit fully connected neural network and $\psi(s,a)$ is a one-layer 256-unit fully connected neural network. Then we apply a one-layer 256-unit fully connected neural network to $\psi(s,a) \odot \varphi(\tau)$.

### C.3  IMPLEMENTATION DETAILS

We implement ORDER based on DSAC and keep the DSAC-specific hyperparameters the same. This hyperparameters are detailed in table 4. As with CODAC, we introduce hyperparameters $\beta, \zeta, \xi$ (See Appendix A.1). In most cases, $\beta$ is a learnable parameter initialized to 1 with learning rate $\eta_\beta = 3 \times 10^{-4}$. In a few cases, setting $\beta = 1$ throughout the entirety of training, which we indicate by setting $\zeta = -1$.

Since we introduce smoothing techniques to the policy and quantile networks, we also add other hyperparameters. In Eq (2), the weight $\alpha$ for the quantile network smoothing loss $\mathcal{L}_{\text{smooth}}$ is searched in $\{0.0, 0.0001\}$. And beyond that, the weight $\alpha_1$ of the policy smoothing loss in Eq. (3) is searched in $\{0.0, 0.1, 1.0\}$. When training the policy and distribution action-value functions, we randomly sample $n = 10$ perturbed observations from a $\ell_\infty$ ball of norm $\epsilon$ and select the one to maximize $D_J(\pi_\theta(\cdot|s)||\pi_\theta(\cdot|\hat{s}))$ and $\mathcal{L}_{\text{smooth}}$, respectively. For $Z$ smoothing loss in Eq. (1), set parameter $\varrho$ to 0.2 for conservative value estimation. All the hyperparameters used in ORDER for the benchmark are listed in Table 5.

Table 4: ORDER backbone hyperparameters

| Hyper-parameter | Value |
|---|---|
| Discount factor $\gamma$ | 0.99 |
| Batch size | 256 |
| Replay buffer size | 1e6 |
| Optimizer | Adam |
| Minimum steps before training | 1e4 |
| Policy network learning rate $\eta_{\text{actor}}$ | 3e-4 |
| Quantile network learning rate $\eta_Z$ | 3e-5 |
| Huber regression threshold $\kappa$ | 1 |
| Number of quantile fractions $N$ | 32 |
| Quantile fraction embedding size | 64 |

Table 5: Hyperparameters of ORDER for the benchmark results

| Datasets | $\xi$ | $\zeta$ | $\eta_{\text{critic}}$ | entropy tuning | $\alpha$ | $\alpha_1$ |
|---|---|---|---|---|---|---|
| hopper-random | 1 | 10 | 3e-5 | yes | 0.0001 | 0.1 |
| hopper-medium | 10 | 10 | 3e-4 | yes | 0.0 | 0.0 |
| hopper-med-rep | 1 | 10 | 3e-5 | yes | 0.0 | 0.0 |
| hopper-med-exp | 10 | 10 | 3e-5 | no | 0.0001 | 0.1 |
| walker2d-random | 1 | 10 | 3e-5 | yes | 0.0001 | 1.0 |
| walker2d-medium | 10 | 10 | 3e-5 | no | 0.0001 | 1.0 |
| walker2d-med-rep | 1 | 10 | 3e-5 | yes | 0.0 | 0.0 |
| walker2d-med-exp | 10 | 10 | 3e-5 | no | 0.0001 | 1.0 |
| halfCheetah-random | 1 | 10 | 3e-5 | yes | 0.0001 | 0.1 |
| halfCheetah-medium | 10 | 10 | 3e-5 | no | 0.0001 | 0.1 |
| halfCheetah-med-rep | 1 | 10 | 3e-5 | yes | 0.0001 | 0.1 |
| halfCheetah-med-exp | 0.1 | -1 | 3e-4 | no | 0.0 | 0.0 |

