# OpenReview forum: "Offline Robustness of Distributional Actor-Critic Ensemble Reinforcement Learning"
_ICLR.cc/2024/Conference — ICLR 2024 Conference Withdrawn Submission_

### Official Review · Reviewer_kZn3 · 2023-10-30

**Soundness:** 2 fair
**Presentation:** 2 fair
**Contribution:** 2 fair
**Rating:** 3
**Confidence:** 3

**Summary:**

This work aims to solve the offline RL problem considering a possibly slightly perturbed environment (e.g., the state observation is noisy). It proposed a new algorithm called ORDER and evaluated its theoretical and empirical performance success in D4RL benchmark.

**Strengths:**

1. The topic of this work is of great interest. The goal is not to consider the performance of a clean environment but can be a slightly perturbed environment w.r.t. state, dynamics, and so on.
2. The background and preliminary is introduced clearly.

**Weaknesses:**

1. The problem formulation of this work is not introduced clearly. It seems there is a lack of introduction or mathematical formulation of what exactly the offline robust RL that this paper targets. The definition of robust RL [1-3] usually refers to the distributionally robust RL against model (transition) perturbation. While the one that this paper considers seems more related to state-adversarial robust RL [4]. The terminology and the real goal of this paper need to be made more clear and explicit.
2. There are five components in the proposed ORDER algorithm, which is also introduced in detail. However, it is hard to see which part is new in ORDER and which dominates the performance in the later experiments, without ablation study of different components as well.
3. The experimental results seem not sufficient since this work only compared the performance of OEDER to that of some basic offline RL algorithms (e.g., CQL, BEAR), while a lot of advanced ones have not been considered, such as IQL.


[1] Moos, Janosch, et al. "Robust reinforcement learning: A review of foundations and recent advances." Machine Learning and Knowledge Extraction 4.1 (2022): 276-315.
[2] Zhou, Z., Bai, Q., Zhou, Z., Qiu, L., Blanchet, J., and Glynn, P. (2021). Finite-sample regret bound for distributionally robust offline tabular reinforcement learning. In International Conference on Artificial Intelligence and Statistics, pages 3331–3339. PMLR.
[3] Shi, Laixi, et al. "The Curious Price of Distributional Robustness in Reinforcement Learning with a Generative Model." arXiv preprint arXiv:2305.16589 (2023).

[4] Zhang, Huan, et al. "Robust deep reinforcement learning against adversarial perturbations on state observations." Advances in Neural Information Processing Systems 33 (2020): 21024-21037.

**Questions:**

1. What is the average/overall performance of ORDER compared to baselines, since there is only per-task performance in Table 1.
2. There are too many details of the experiments in Section 5.2, which may be better to leave to the appendix and leave space for a more comprehensive analysis/evaluation of the proposed method ORDER.
3. The setting of the experiments seems not clear to the reviewer. As the goal of this work is to handle possible state perturbation, should the experiments be conducted in some settings where the state is perturbed?

---

> ### Author Response · Authors · 2023-11-22
> **Answer to Reviewer kZn3 1**
>
> *``What is the average/overall performance of ORDER compared to baselines, since there is only per-task performance in Table 1. ”*
>
> **Answer.** Thanks for your suggestion. We have added the average performance of ORDER in Table 1 in the new version. Meanwhile, the average performance of ORDER surpasses most basic offline RL algorithms.
>
> *``There are too many details of the experiments in Section 5.2, which may be better to leave to the appendix and leave space for a more comprehensive analysis/evaluation of the proposed method ORDER. ”*
>
>
> **Answer.** Thank you for pointing this issue. We will add the results of ORDER's experiments in more environments in Section 5.2 for better analysis.
>
> *``The setting of the experiments seems not clear to the reviewer. As the goal of this work is to handle possible state perturbation, should the experiments be conducted in some settings where the state is perturbed? ”*
>
> **Answer.** Thanks for your valuable suggestion. In further work, we will supplement the ablation study of ORDER performance with varying perturbation scale.

---

### Official Review · Reviewer_fvaB · 2023-10-31

**Soundness:** 2 fair
**Presentation:** 3 good
**Contribution:** 2 fair
**Rating:** 3
**Confidence:** 4

**Summary:**

This paper introduces an offline RL algorithm ORDER that leverages distributional RL to learn a policy robust to the distribution shift. Besides the conventional distributional RL loss, the proposed method adopts an ensemble of quantile networks, introduces smoothness regularizers for both policy and value functions, and penalizes the value of OOD actions. The authors further conduct experiments on D4RL to compare the proposed methods with baselines.

**Strengths:**

1. This paper empirically evaluates the proposed method on 12 tasks from the Gym-MuJoCo domain of the D4RL benchmark.

2. This paper provides some theoretical analysis of the proposed method.

**Weaknesses:**

1. The proposed method lacks enough novelty. First, as mentioned by the authors, the techniques of distributional RL have already been introduced into the offline RL settings. Second, adopting an ensemble of value networks is a widely known technique that can improve performance in the RL community. Third, adding smoothness constraints in the offline RL settings has already been explored in the offline RL settings [1].

2. The ablation studies are weak. I suggest the authors investigate how different regularizing coefficients (e.g., $\alpha$, $\beta$) impact the final performance.

3. The authors claim existing distributional offline RL algorithms "leverage a conservative return distribution to impair the robustness, and will make policies highly sensitive" without providing enough empirical support. To me, I cannot see why leveraging a conservative return distribution can impair the robustness of the learned policy.

4. The authors miss a related work [2], which also leverages distributional RL to learn value functions. I suggest the authors include [2] as another baseline method in Table 1.

[1] Sinha et al., S4RL: Surprisingly Simple Self-Supervision for Offline Reinforcement Learning in Robotics. 5th Annual Conference on Robot Learning (CoRL)

[2] Li et al., Offline Reinforcement Learning with Closed-Form Policy Improvement Operators. ICML 2023.

**Questions:**

See the Weaknesses section.

---

> ### Author Response · Authors · 2023-11-22
> **Answer to Reviewer fvaB 1**
>
> *``The proposed method lacks enough novelty. First, as mentioned by the authors, the techniques of distributional RL have already been introduced into the offline RL settings. Second, adopting an ensemble of value networks is a widely known technique that can improve performance in the RL community. Third, adding smoothness constraints in the offline RL settings has already been explored in the offline RL settings. ”*
>
> **Answer.** First, distributional RL has been applied to offline RL [1,2]. However, it is well known that distributional RL is widely used in online setting while its academic exploration is still incomplete in offline setting. Existing distributional offline RL methods only focus on the safety of the learned policy. These methods leverage a conservative return distribution to impair the robustness. In contrast, our work considers improving the robustness of the learned policies under control of the distribution shift.
>
> Second, adopting an ensemble of value networks is a widely known technique that can improve performance in the RL community. On the one hand, double $Q$ networks are often used in RL algorithms to alleviate the overestimation of $Q$ functions. On the other hand, the number of distributed integrated networks has been explored in online setting [3]. However, offline RL is different from online reinforcement learning in that it has some additional problems, such as distribution shift and OOD problems. Many scholars' hasty use of the double network technique has hindered the exploration of the number of value networks in ensemble in the field of offline RL.
>
> Third, it is known that adding smoothness constraints to offline RL has been explored in the offline settings, such as S4RL [4] and RORL [5]. However, some previous related work enforced smoothness in $Q$ functions, and the smoothing treatment in our work applied to distribution action value functions. We can use the framework of distributional RL to consider risk-sensitive learning and the performance of the algorithm under different risk measures, so as to ensure the security of learned polices. Besides, when the $Q$ value is updated, the loss function takes the form of mean square error. Huber loss is adopted in distributional RL, which can better fit our research. Therefore, the problem we studied is an interesting and meaningful work.
>
> [1]Will Dabney, et al., Implicit quantile networks for distributional reinforcement learning. In International Conference on Machine Learning, pp. 1096-1105, 2018a.
>
> [2] Xiaoteng Ma, et al., DSAC: Distributional soft actor critic for risk-sensitive reinforcement learning. arXiv preprint arXiv:2004.14547, 2020.
>
> [3] Lan, Q., et al., Maxmin q-learning: Controlling the estimation bias of q-learning. In 8th International Conference on Learning Representations, ICLR 2020, Addis Ababa, Ethiopia, April 26-30, 2020.
> [4] Sinha et al., S4RL: Surprisingly Simple Self-Supervision for Offline Reinforcement Learning in Robotics. 5th Annual Conference on Robot Learning (CoRL).
>
> [5] Rui Yang, et al., RORL: Robust
> offline reinforcement learning via conservative smoothing. In Advances in Neural Information
> Processing Systems, volume 35, pp. 23851–23866, 2022.

---

> > ### Author Response · Authors · 2023-11-22
> > **Answer to Reviewer fvaB 2**
> >
> > *``The ablation studies are weak. I suggest the authors investigate how different regularizing coefficients (e.g., $\alpha,\beta$) impact the final performance. ”*
> >
> > **Answer.** Thanks for your valuable suggestion. We will complement ablation experiments for different regularizing coefficients in the subsequent work. The results verify that the value of $\alpha,\beta,\alpha_{1}$ in our experiments (Section 5) exhibits a well performance.
> >
> > *``The authors claim existing distributional offline RL algorithms "leverage a conservative return distribution to impair the robustness, and will make policies highly sensitive" without providing enough empirical support. To me, I cannot see why leveraging a conservative return distribution can impair the robustness of the learned policy. ”*
> >
> > **Answer.** As we all know, CQL [1] is a classic conservative offline reinforcement learning algorithm. In RORL [2], the authors demonstrate that current value-based offline RL algorithms lack the necessary smoothness for the policy. A motivating example is presented to illustrate the robustness of the well-known baseline method CQL policy, where even a tiny scale perturbation on observation can cause significant performance degradation. Therefore, leveraging a conservative return distribution can impair the robustness of the learned policy.
> >
> > *``The authors miss a related work, which also leverages distributional RL to learn value functions. I suggest the authors include as another baseline method in Table 1. ”*
> >
> > **Answer.** Thank you very much for pointing this out. We have added the result of this work to Table 1 in the new version. Significantly, our algorithm surpasses the performance of this work.
> >
> > [1] Aviral Kumar, et al., Conservative q-learning for offline reinforcement learning. In Advances in Neural Information Processing Systems, volume 33, pp.1179-1191, 2020.
> >
> > [2] Rui Yang, et al., RORL: Robust offline reinforcement learning via conservative smoothing. In Advances in Neural Information
> > Processing Systems, volume 35, pp. 23851–23866, 2022.

---

### Official Review · Reviewer_1eAG · 2023-10-31

**Soundness:** 3 good
**Presentation:** 2 fair
**Contribution:** 2 fair
**Rating:** 3
**Confidence:** 3

**Summary:**

This paper delves into the realm of offline robust reinforcement learning, introducing the Offline Robustness of Distributional Ensemble Reinforcement Learning (ORDER) as its primary contribution. ORDER aims to strike a balance between conservatism and robustness in offline settings, utilizing an ensemble of multiple quantile networks to enhance its resilience. Notably, it incorporates a smoothing technique for policies and distributional functions, primarily focusing on perturbed states. The paper reinforces its claims with a theoretical proof of ORDER's convergence to a conservative lower bound, substantiated through empirical experiments.

**Strengths:**

- The utilization of an ensemble of multiple quantile networks holds promise in bolstering the approach's robustness.

- The inclusion of a theoretical proof demonstrating convergence to a conservative lower bound adds weight to ORDER's efficacy in enhancing robustness.

**Weaknesses:**

- While the paper introduces ORDER as a modification of the prior work RORL into a distributional version, it is essential to engage in a comprehensive discussion comparing this work with RORL to underscore its significance and contributions more effectively.

- The experiments presented in Table 1 do not convincingly showcase the superiority of ORDER over the previous state-of-the-art offline RL algorithm, RORL. This raises concerns about the relevance and significance of the studied problem: offline distributional RL.

- The ablation experiments, considering the number of introduced components, are rather limited. For a more comprehensive understanding, it would be valuable to explore the individual impacts of components like the policy smooth loss, OOD penalty, and state perturbation on the proposed algorithm's performance.

In summary, it is crucial for this work to accentuate the importance of the problem it addresses and the uniqueness of the proposed method. In the realm of sophisticated algorithms, significant advantages are generally expected. Additionally, a more comprehensive set of ablation experiments would further strengthen the paper.

**Questions:**

Most questions are listed in the weakness part. There are also some typos and notations need to be defined:

- The parentheses in the definition of $Q^\pi$ are incorrectly positioned.

- In the definition of $\hat{\pi}$: Define and unify the notation of the indicator function $\mathbf{1}$.

- Define the notation $\mathcal{P}(\mathbb{R})$ in Section 2.2.

- Define the notation $\mathbb{B}_d(s, \epsilon)$.

---

> ### Author Response · Authors · 2023-11-22
> **Answer to Reviewer 1eAG 1**
>
> *`` There are also some typos and notations need to be defined: ”*
>
> *``The parentheses in the definition of $Q^{\pi}$ are incorrectly positioned.  ”*
>
> *``In the definition of $\hat{\pi}$: Define and unify the notation of the indicator function ${1}$. ”*
>
> *``Define the notation $\mathcal{P}(\mathbb{R})$ in Section 2.2. ”*
>
> *``Define the notation $\mathbb{B}_{d}(s,\epsilon)$. ”*
>
> **Answer.** Thanks for your suggestion. We have added the definitions of the indicator function ${1}$ and the notation $\mathcal{P}(\mathbb{R})$. We also have modified the pasentheses in the definition of $Q^{\pi}$ and the inconsistency of the indicator function ${1}$. In fact, the definition of $\mathbb{B}_{d}(s,\epsilon)$ has already been given in Section 3.1.
>
> *``While the paper introduces ORDER as a modification of the prior work RORL into a distributional version, it is essential to engage in a comprehensive discussion comparing this work with RORL to underscore its significance and contributions more effectively. ”*
>
> *``The experiments presented in Table 1 do not convincingly showcase the superiority of ORDER over the previous state-of-the-art offline RL algorithm, RORL. This raises concerns about the relevance and significance of the studied problem: offline distributional RL. ”*
>
> *``The ablation experiments, considering the number of introduced components, are rather limited. For a more comprehensive understanding, it would be valuable to explore the individual impacts of components like the policy smooth loss, OOD penalty, and state perturbation on the proposed algorithm's performance. ”*
>
> **Answer.** Thank you for pointing this out. You may have some misconceptions about what we do. In this paper, we introduce smoothing techniques from RORL into CODAC to enhance the robustness of offline distributional reinforcement learning. In our experiments, the performance of ORDER surpasses offline distributional reinforcement learning algorithms, such as CODAC, MQN-CQR. Meanwhile, ORDER also outperformed RORL results on datasets such as hopper-medium-replay and walker2d-random.

---

### Official Review · Reviewer_cjFe · 2023-11-01

**Soundness:** 2 fair
**Presentation:** 2 fair
**Contribution:** 2 fair
**Rating:** 5
**Confidence:** 3

**Summary:**

The paper proposes the algorithm ORDER (Offline Robustness of Distributional actor-critic Ensemble Reinforcement learning) to improve the robustness of policies in offline reinforcement learning (RL) settings. ORDER introduces two approaches to enhance robustness: i) smoothing technique to policies and distribution functions for states near the dataset, and ii) strengthening the quantile network. The algorithm incorporates a dynamic entropy regularizer of the quantile function to ensure sufficient exploration and controls the distribution shift. The paper theoretically proves that ORDER converges to a conservative lower bound, which helps alleviate the distribution shift. Experimental validation on the D4RL benchmark demonstrates the effectiveness of ORDER in improving policy robustness.

**Strengths:**

- The paper introduces the algorithm ORDER, which addresses the challenges of distribution shift and robustness in offline RL settings.
- The paper provides theoretical proofs of the convergence of ORDER to a conservative lower bound, which helps alleviate the distribution shift.
- The paper clearly presents the algorithm ORDER and its components, including the smoothing technique, strengthening the quantile network, and dynamic entropy regularizer.

**Weaknesses:**

- The paper lacks a thorough analysis of the computational complexity and scalability of the ORDER algorithm, which could be important considerations for real-world applications. As the distributional RL needs more computational resources.

- The experimental validation of ORDER is limited to the D4RL benchmark, and it would be beneficial to evaluate the algorithm on a wider range of tasks and datasets to demonstrate its generalizability.

- RORL seems better than ORDER.

**Questions:**

- Could the authors provide insights into the computational complexity and scalability of the ORDER algorithm?
- It would be beneficial to evaluate the ORDER algorithm on a wider range of tasks and datasets beyond the D4RL benchmark. This would demonstrate the generalizability of ORDER and its effectiveness in different domains.

**Details Of Ethics Concerns:**

No ethics concerns.

---

> ### Author Response · Authors · 2023-11-22
> **Answer to Reviewer cjFe 1**
>
> *``Could the authors provide insights into the computational complexity and scalability of the ORDER algorithm? ”*
>
> **Answer.** On the one hand, for some offline reinforcement learning algorithms, such as RORL [1], our algorithm is to model quantile networks of action values which will introduce more network parameters. On the other hand, for offline distributional reinforcement learning algorithms such as CODAC [2], we add a smooth term loss. Although these will increase the computational complexity to some extent, the performance of ORDER is much higher than the current advanced offline distributional reinforcement learning algorithms, such as CODAC, MQN-CQR. Meanwhile, ORDER also outperforms RORL results on datasets such as hopper-medium-replay and walker2d-random.
>
> *``It would be beneficial to evaluate the ORDER algorithm on a wider range of tasks and datasets beyond the D4RL benchmark. This would demonstrate the generalizability of ORDER and its effectiveness in different domains. ”*
>
> **Answer.** Thanks for your comment. The wider application of our algorithm in real-world scenarios will be a future work.
>
> [1] Rui Yang, et al., RORL: Robust offline reinforcement learning via conservative smoothing. In Advances in Neural Information Processing Systems, volume 35, pp. 23851-23866, 2022.
>
> [2] Yecheng Ma, et al., Conservative offline distributional reinforcement learning. In Advances in Neural Information Processing Systems, volume 34, pp. 19235-19247, 2021.